# Quasi-Synchronous Variations in the OLR of NOAA and Ionospheric Ne of CSES of Three Earthquakes in Xinjiang, January 2020

**Chen Yu [1], Jing Cui [2,*], Wanchun Zhang [3], Weiyu Ma [1], Jing Ren [1], Bo Su [4] and Jianping Huang [2]**

[1]    China Earthquake Networks Center, Beijing 100045, China; yuchen@seis.ac.cn (C.Y.); maweiyu@seis.ac.cn (W.M.); renjing@seis.ac.cn (J.R.)
[2]    National Institute of Natural Hazards, MEMC, Beijing 100085, China; jianpinghuang@ninhm.ac.cn
[3]    National Satellite Meteorological Center, China Meteorological Administration, Beijing 100081, China; zhangwc@cma.gov.cn
[4]    School of Environmental Science and Tourism, Nanyang Normal University, Nanyang 473061, China; subohappy@nynu.edu.cn
[*]    Correspondence: jingcui@ninhm.ac.cn

**Abstract:** The successive tidal force (TF) at the epicenter of the Jiashi M6.6 earthquake in Xinjiang, China, was calculated for the period from 13 December 2019 to 10 February 2020. With periodic changes in tide-generating forces, the variations in the electron density (Ne) data recorded by the China Seismo-Electromagnetic Satellite (CSES) and outgoing longwave radiation (OLR) data provided by NOAA on a large scale at N25°–N55°, E65°–E135° were studied. The results show that (1) in the four cycles during which the TF changes from trough to peak, the earthquake occurred during one peak time when the OLR changed around the epicenter via calm–rise processions and in other similar TF phases, and neither an increase in the OLR nor earthquake occurred. (2) With a change in the TF, the spatiotemporal evolution of the OLR from seismogenic processes to its occurrence was as follows: microenhancement–enhancement–microattenuation–enhancement–calmness; this is consistent with the evolution of outward infrared radiation when rocks break under stress loading: microrupture–rupture–locking–accelerated rupture–rupture. (3) Ne increased significantly during the seismogenic period and was basically consistent with OLR enhancement. The results indicate that as the TF increases, the Earth's stress accumulates at a critical point, and the OLR increases and transfers upward. The theoretical hypothesis underlying the conducted study is that the accumulated electrons on the surface cause negatively charged electrons in the atmosphere to move upward, resulting in an increase in ionospheric Ne near the epicenter, which reveals the homology of seismic stress variations in the spatial coupling process. The quasi-synchronous change process of these three factors suggests that the TF changed the process of the stress accumulation–imbalance in the interior structure of this earthquake and has the effect of triggering the earthquake, and the spatiotemporal variations in the OLR and ionospheric Ne could be indirect reflections of in situ stress.

**Keywords:** Jiashi earthquake; OLR; CSES; electron density (Ne)

## 1. Introduction

In the 1980s, Gorny et al. [1] discovered a large area of thermal infrared satellite radiation enhancement anomalies (10.5–12.5 μm) before the Gazli earthquake in Central Asia. Ouzounov et al. [2] and Qin et al. [3] have used outgoing longwave radiation (OLR) data and the image difference method to comprehensively study infrared phenomena related to major earthquakes (26 January 2001: M7.9 Bhuj Gujarat India earthquake; 21 May 2003: M6.8 Boumerdes North Algeria earthquake; 26 December 2003: M6.6 Southeastern Iran earthquake; 26 December 2004: M9.0 Sumatra–Andaman earthquake; 3 September 2010: M 7.1 New Zealand earthquake). They considered that there were obvious anomalies

within 20 d before the earthquake, and the anomaly's amplitude was 6–80 w/m$^2$. Kong used the geometric moving average martingale method to process historical OLR data and showed that there was an OLR anomaly before the Wenchuan earthquake [4]. Recently, pre-earthquake thermal anomalies have also been verified in many earthquake cases [5–7]. The use of thermal infrared remote sensing for monitoring pre-earthquake anomalies at a broadly spatial and continuous temporal scale has become a popular topic in seismic monitoring research [8].

Currently, statistical methods are often used to extract seismic anomalies. Common algorithms include robust satellite techniques [9,10], the eddy field calculation mean algorithm [11,12], the spatiotemporally weighted two-step method, and climatological analysis for seismic precursor identification [13]. These methods require many years of historical data. The selection of background data primarily relies on multi-year mean values or mean-related algorithms, lacking a background date construction algorithm with physical significance. The abnormal results exhibit several issues: large anomaly area distributions [14]; the high dispersion degree of the anomaly range's distribution [15]; the abnormal time is polycyclic [16]; and the results are ambiguous [17–19]. Therefore, it is necessary to develop a calculation method for the background time index with clear physical significance and obtain the infrared anomalies.

An earthquake is a process during which the Earth's tectonic stress accumulates to a certain intensity, breaking through the critical value of rock elastic rupture and releasing energy quickly. Previous studies have shown that thermal infrared radiation (TIR) changes relative to stress in loaded rocks [20]. The results of mechanical rock loading and fracture tests show that the infrared radiation generated during rock failure exhibits a time series phase change with respect to microintensification–intensification–attenuation–intensification–quietness synchronization [21–23], which provides a physical basis, monitoring content, and abnormal evaluation criteria for infrared remote sensing earthquake monitoring and prediction. Tidal force (TF) is an essential external mechanical factor that triggers earthquakes when crustal stress reaches a critical state [24,25]. It is also the only physical parameter of earth deformations that can be calculated in advance and has an indicative advantage over time [26]. Therefore, the extraction of seismic thermal anomalies based on tidal forces is helpful for obtaining infrared anomalies that conform to the radiation variation law of rock stress rupture as it can improve the recognition level of seismic infrared anomalies in time and space.

A previous study recorded the disturbance phenomenon of ionospheric D, E, and F layers in the vicinity of earthquakes [27–33]. Parrot et al. found that the electron density (Ne) over the Kii Peninsula increased significantly during the 7 d before the earthquake [33]. Sarkar et al. studied the abnormal evolution process of Ne before three earthquakes in the mid-latitude region and found that the maximum variation amplitude was 20% [34], and other seismologists [35,36] have found that the ionospheric electron density and ion density also increased before the 2008 Wenchuan 8.0 earthquake. Based on these examples, a conceptual model of ring-layer coupling has been proposed [37,38]. However, based on the ring-coupling mechanism, there remains a lack of corresponding synchronous research on the effect of the OLR on the physical parameters of the ionosphere at high altitudes, such as ionospheric electron density (Ne). The successful launch of the China Seismo-Electromagnetic Satellite (CSES) on 2 February 2018 made possible the further analysis of the spatial distribution of OLR anomalies, the energy transfer of electromagnetic radiation, and the spatial coupling relationship of multiparameter spatial anomalies.

In this study, based on the period of TF variations, the continuous diurnal variation tracking of multiple parameters during the Jiashi earthquake was conducted simultaneously with OLR and Ne observation data, excluding the interference of magnetic storms. This study can further confirm that the thermal anomalies based on the tidal force method are related to earthquakes.

## 2. Tectonic Environment of Jiashi Earthquake

On 18 January 2020, an Ms 5.4 earthquake (N39.83°, E77.18°, focal depth of 20 km) occurred in Jiashi County, Kashgar Prefecture, Xinjiang Autonomous Region (http://www.cenc.ac.cn/cenc/dzxx/359712/index.html (accessed on 24 November 2023)); on 19 January 2020, this was followed by another Ms6.4 earthquake (N39.83°, E77.21°, focal depth of 16 km) in Jashi County (http://www.cenc.ac.cn/cenc/dzxx/359697/index.html (accessed on 24 November 2023) and an Ms5.2 earthquake in Atushi City, Xinjiang (N39.89°, E77.46°, focal depth of 14 km) (http://www.cenc.ac.cn/cenc/dzxx/359712/index.html (accessed on 24 November 2023). The Jiashi earthquake occurred within the annual M 6.0 ± 0.2 potential earthquake risk region of western Xinjiang in 2020. It was predicted by the China Earthquake Networks Center in November 2019, which used thermal infrared remote sensing technology (atmospheric temperature and OLR parameters); the monthly M 5.2 ± 0.3 potential earthquake risk region was predicted during a consultation that took place on 12 December 2019 (Figure 1) [39].

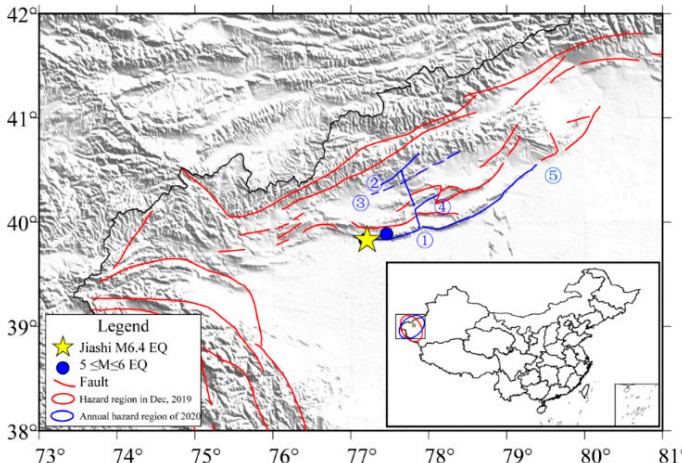

**Figure 1.** Distribution of structure and prediction areas in Xinjiang. Note: 1. PQ fault; 2. AYBLK fault; 3. KBKSS fault; 4. TTAETG fault; 5. KPTG fault.

From the perspective of the tectonic environment, the earthquake mainly occurred in the fold-reverse fault structural belt on the front (south) edge of the Kepingtage (KPTG) fault [40]. The KPTG fault is 300 km long in the east–west direction and 65–75 km wide in the north–south direction. It comprises many rows of nearly parallel NEE-trending fold belts in the horizontal direction, including several groups of inverted compound anticlines, compound box anticlines, and other complex structures. The NE-trending Piqiang (PQ) fault is divided into east and west parts. Vertically, five rows of thrust fold belts are located west of the PQ fault and situated from the north to the south, including Aoyibulake (AYBLK), Kebukesanshan (KBKSS), and KPTG. Paleozoic and Cenozoic strata are exposed westward from the PQ fault, forming a nose-like structure that dips westward. From north to south to the east of the PQ fault, there are six rows of thrust fold belts, including the AYBLK, KBKSS, PQ, Tataaiertage (TTAETG), and KPTG fault [41]. Several strong earthquake swarms with magnitudes of six in 1997 and 2003 occurred near the epicenter, and the tectonic activity of the area was strong.

## 3. Data and Method

### 3.1. Data Selection

To obtain the spatial variation characteristics of radiation and considering that the outward radiation band of Earth is mainly concentrated in the longwave band, we used OLR data [42]—which most directly reflects the underlying surface properties, energy variation parameters, and atmospheric window band—as the research object. To ensure the continuity and universality of data, we adopted the ground-emitted longwave radiation products

of the NOAA satellite, with a spatial resolution of $1° \times 1°$, covering a total of $360 \times 181$ grid points worldwide and a temporal resolution of 1 d. To test the anomaly monitoring efficiency of the tidal background selection method for seismic anomaly identification, we considered a large area (N25°–N55°, E65°–E135°) as the research object. The data are available from the following website: https://ftp.cpc.ncep.noaa.gov/-precip/noaa18_1x1/ (accessed on 5 January 2021).

The ionospheric electron density (Ne) in an ascending model (local daytime) was obtained from the Langmuir Probe of the level 2 product of CSES-01. CSES-01 is a sun-synchronous satellite orbiting at a height of approximately 507 km with a 02:00 local time ascending node. CSES-01 carries eight types of scientific payloads: high-precision magnetometers, electric field detectors, search coil magnetometers, Langmuir probe (LAP), plasma analyzer packages, global navigation satellite system (GNSS) occultation receivers, three frequency beacons, and energetic particle detectors. Among these payloads, the LAP is the payload for the in situ detection of space plasma, which can obtain electron densities, and the relevant design parameters can be found in [43–45]. The working mode of the LAP comprises survey and burst modes: the former is mainly used to detect the global electron density with a sweeping period of 3 s, and the latter is mainly used to detect the sky over China, major global seismic zones, and other areas of focus with a sweeping period of 1.5 s. The data utilized in this paper are available from the following website: https://www.leos.ac.cn/ (accessed on 14 January 2021).

*3.2. Method*

The stress in the Earth's crust results from both tectonic and tidal forces acting upon it. When seismic tectonic stress reaches a critical state, causing rock sliding, and if rapidly changing tidal stress is superimposed relative to an appropriate direction, it may trigger an earthquake Heaton, 1975 [24,46,47]. The tidal force is periodically and continuously changing, and the phase of its seismic induction varies with respect to seismic structure [48]. In this study, the tidal force fluctuant analysis method was used for thermal anomaly extraction. It contains two steps: The first step is to calculate the change curve of the tidal forces in the epicenter over time. The tidal force produced by the Sun and Moon at the epicenter is related to the zenith distance, the mass of the celestial body, the distance between the epicenter and the center of the Earth, etc. The tidal force generated by any celestial body $\varphi$ at a point inside the Earth is $W_\varphi(P)$. It can be calculated using Equation (1). In Equation (1), $P_n(\cos z_m)$ is a Legendre polynomial; $z_m$ is the star's zenith distance; M is the mass of the celestial body; k is the universal gravitation constant; r is the distance between the epicenter and the center of the Earth; $r_m$ is the distance between the center of mass and the center of the Earth. According to Equation (1), the tidal force generated by the Sun and the Moon is calculated, and the last two tidal forces are superimposed to obtain the solar and lunar tidal force of the epicenter [49].

The second step is OLR anomaly extraction. Based on the tidal cycle from step 1, the starting time of each cycle was taken as the reference time background, and the change in satellite OLR data was extracted according to Equation (2). In Equation (2), $\Delta OLR_i(x, y)$ is the OLR incremental value of each grid point, $OLR_i(x, y)$ is the OLR value of each grid point, and $OLR_{background}(x, y)$ is the OLR value of the background.

$$W_\varphi(p) = k\frac{M}{r_m}\sum_{n=2}^{\infty}\left(\frac{r}{r_m}\right)^n P_n(\cos z_m) \tag{1}$$

$$\Delta OLR_i(x, y) = OLR_i(x, y) - OLR_{background}(x, y) \tag{2}$$

Ne data were also selected (N25°–N55°, E65°–E135°). The overhead time in the range is approximately 16–21 h local time, which ensures that there are 3–4 tracks every day, and the distance between each track is approximately 500 km.

Above all, original ΔOLR and Ne data were analyzed simultaneously, also taking into account the variations in the tidal force (TF). The results are presented in the form of three-layered 2D latitudinal–longitudinal graphical schemes.

## 4. Results

### 4.1. Tidal Force Change

For this earthquake, we used the TF method to calculate the continuous change in the TF from 13 December 2019 to 10 February 2020 at the epicenter and drew the curve, as shown in Figure 2. The abscissa is the time series, and the ordinate is the tidal force (Gal). The image shows that the tide-generating force experienced four continuous cycles (A, B, C, and D) from trough to peak to trough. In this article, we use the starting time of each cycle as the background time for the following cycle, and x, y, and i are used to denote the latitude, longitude, and grid-point mark, respectively. Previous studies have shown that the phase of the tidal force's position at the time of the earthquake has a certain relationship with the type of earthquake. Normal fault and strike–slip-type earthquakes mostly occur near the peak of the tidal force's position. Usually, the first low-value phase moment before these phenomena is selected as the background moment; then, the moment is subtracted daily. Reverse fault-type earthquakes mostly occur in the low-tide position, and the preseismic high point is usually chosen as the background of this period [19,50,51].

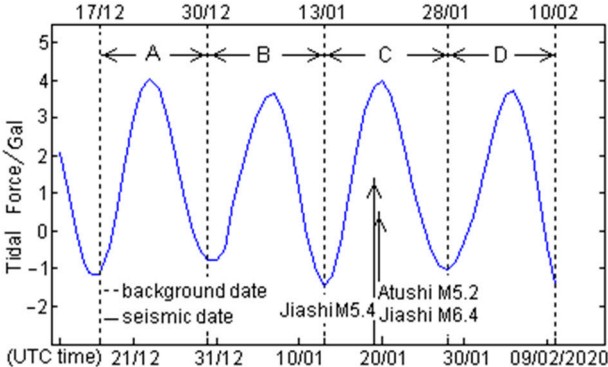

**Figure 2.** Temporal variation diagram of the tide force of Jiashi earthquake in Xinjiang. A, B, C and D are the cycle of TF variation.

During this earthquake, for each cycle of TF variation (A, B, C, and D), nighttime surface longwave data (OLR) on 17 December 2019, 30 December 2019, 13 January 2020, and 28 January 2020 were selected as the background (the turning point of each cycle is the point closest to the low point). The OLR within the spatial range (N25°–N55°, E65°–E135°) was calculated from 18 December 2019 to 10 February 2020. Based on Equation (2), the daily continuous change images of the OLR in each stage before and after the earthquake were obtained, which can be used as the basis for the analysis of the impending radiation change in the earthquake.

### 4.2. Space Weather Background

The ionospheric pattern is closely connected to solar activity, which can be reflected by the Dst and Kp index. In general, the Kp and Dst index can be used to determine whether a disturbance is caused by a space event or an earthquake. If the Kp index is less than or equal to 30 or the Dst index is greater than or equal to –30 nT, then the period is considered to be quite geomagnetically calm. Figure 3 shows the indexes from 13 December 2019 to 10 February 2020, covering the range from 36 days before the earthquake to 22 days after the earthquake. Figure 3 shows that the period is geomagnetically calm (https://wdc.kugi.kyoto-u.ac.jp/ (accessed on 12 December 2021)).

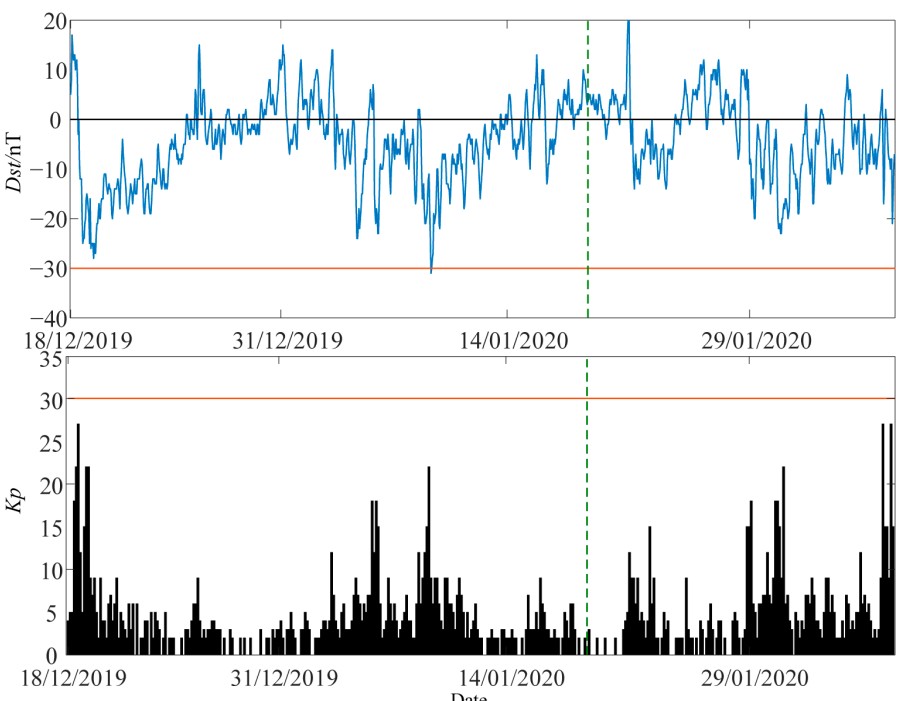

**Figure 3.** Space weather indices. The image above is the Dst index, and the image below is the Kp index. The red line in each subplot denotes the boundary for checking whether there is a space weather event or not, and the vertical green line is the time of the earthquake.

*4.3. Temporal and Spatial Change Characteristics of the OLR and Ne*

To improve the comparison of the anomalous evolution of the OLR and Ne before the earthquake, we plotted the same-day data of the two. As shown in Figures 4–7 among the four tidal cycles A, B, C, and D, an earthquake occurred only in cycle C, which was also the only period in which the OLR was significantly enhanced. In the entire country, the OLR changed significantly before and after the earthquake only near the epicenter and its vicinity: On 14 January, a small area of weak radiation enhancement appeared. On 15 January, anomalies in the middle east, west, and south of the epicenter had extended, obviously increasing up to 82 W/m$^2$. On 16 January, OLR anomalous diffusion to the southeast was observed. On 17 January, the abnormal OLR began to further strengthen and increase to 84 w/m$^2$. The abnormal OLR remained almost unchanged on 18 January. On 19 January, radiation was enhanced, and it was maintained at 84 W/m$^2$ on the same day of the earthquake. On 20 January, affected by the earthquake, the surface ruptured, resulting in a large amount of surface thermal radiation. The OLR anomaly quickly disappeared after the earthquake on 21 and 22 January.

These results showed that the evolution of the OLR experienced an evolutionary process comprising initial warming → strengthening → peak → continuous weak attenuation → calm during the earthquake, which is similar to the process of some earthquakes [19]. The OLR anomaly was located to the right of the epicenter and presented an isolated cluster distribution in space. The anomaly was distributed along the southern margin of the West Kunlun fault (F1) and the western segment of the South Tianshan Mountains (F2), and it exhibited a high identification degree, further indicating that the anomaly originated from the surface area near the epicenter. According to the analysis of the variation process of the OLR before the earthquake, the process of the OLR before the earthquake is consistent with the evolution characteristics of rock stress fracture in space: compression → micro-fracture → fracture extension → stress locking → rupture termination [21]. This phenomenon reveals the process of this earthquake: the time evolution process of rocks from tectonic stress loading → quasi-static nucleation → dynamic rupture → stress redistribution → fault strength recovery [52].

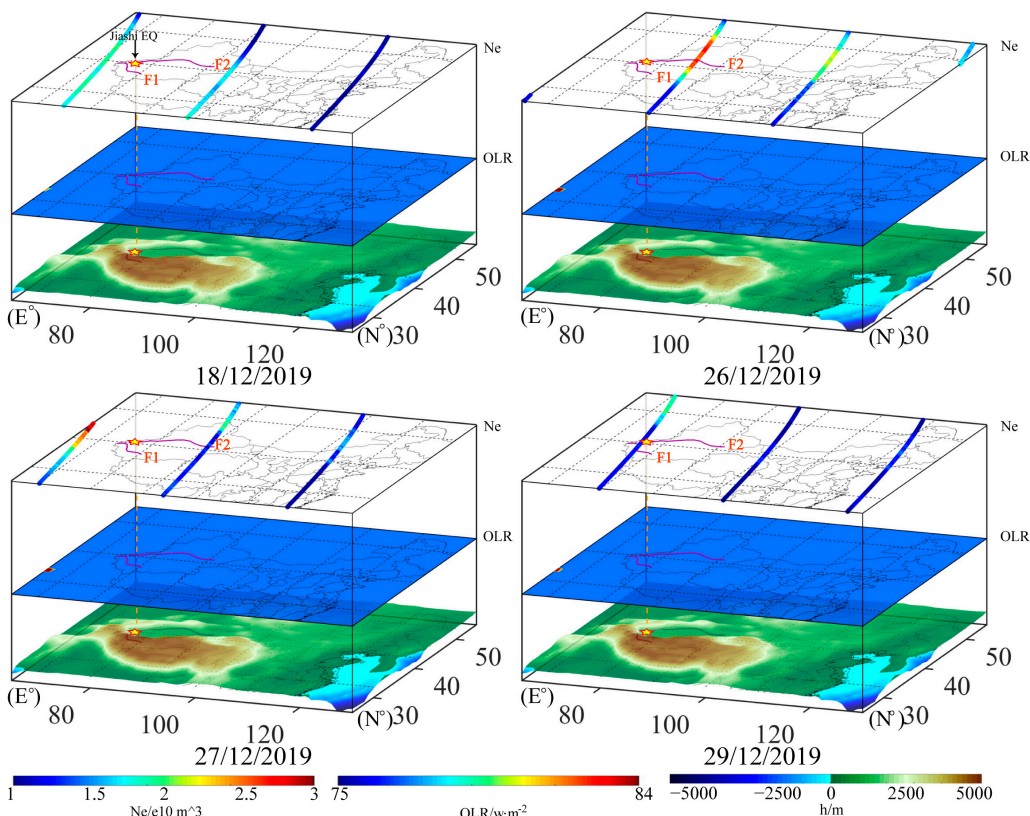

**Figure 4.** Temporal and spatial evolution of the OLR and Ne at the epicenter of the Jiashi earthquake (period A). There are no OLR anomalies around the epicenter. Note: The top layer is the CSES orbit map of Ne, the middle layer is the distribution map of the OLR, the bottom layer is the digital elevation model (DEM), the pentagon is the location of the epicenter, F1 is the West Kunlun fault, and F2 denotes the South Tianshan Mountains. The red stars in the top and bottom layers represent the earthquake's epicenter, the vertical red dashed lines link the epicenters of different layers.

The evolution of ionospheric Ne and OLR during the same period within this region exhibits apparent quasi-synchronous characteristics before and after the earthquake. As shown in Figures 4–7 in the four tidal cycles A, B, C, and D, the highest amount of Ne in orbit and closest to the epicenter and latitude range of 25° N–55° N was observed during cycle C of the earthquake, and it was significantly higher than the average value of other periods in the four cycles. During period A (18 December 2019–29 December 2019), the regional Ne in the orbit closest to the epicenter (40° N–50° N) on December 26 and 27 appeared to be the maximum value. The distance between the two orbits from the epicenter was approximately 1000 km and 850 km, reaching $2.79 \times 10^{10}$ cm$^{-3}$. In addition, the average Ne was approximately $1.5 \times 10^{10}$ cm$^{-3}$; during the B period (31 December 2019–11 January 2020), the daily change in Ne in this region was relatively stable, and no significant change was observed and recorded, with an average value of approximately $1.2 \times 10^{10}$ cm$^{-3}$. During the C period (14 January 2020–22 January 2020), the Jiashi earthquake occurred, and Ne exhibited a quasi-synchronous change that was similar to that of OLR. On 14 January, a slight radiation enhancement was observed near the epicenter, with an increase of 82 W/m$^2$. At the same time, the orbital (approximately 300 km away from the epicenter) to the east of the epicenter also exhibited a synchronous increase in Ne, and a positive anomaly occurred in the area that was concentrated at 40° N–45° N. The maximum value reached $3.08 \times 10^{10}$ cm$^{-3}$, and the average value was more than twice that of periods A and B, reaching $2.5 \times 10^{10}$ cm$^{-3}$. On 17 January, the OLR anomaly was further enhanced with an increase of 84 W/m$^2$, and Ne also reached a maximum value of $2.91 \times 10^{10}$ cm$^{-3}$ before the earthquake. The average value within

the 17 January orbit range was further increased compared with the value from 14 January, reaching $2.45 \times 10^{10}$ cm$^{-3}$. After the 20 January earthquake, both the anomalous intensity and anomalous range of Ne and the OLR were attenuated and compared with values before the earthquake. The maximum value of Ne decreased to approximately $2.45 \times 10^{10}$ cm$^{-3}$, and the average value decreased to approximately $2.03 \times 10^{10}$ cm$^{-3}$; period D (29 January 2020–6 February 2020) was similar to period B, and there was no significant change in Ne.

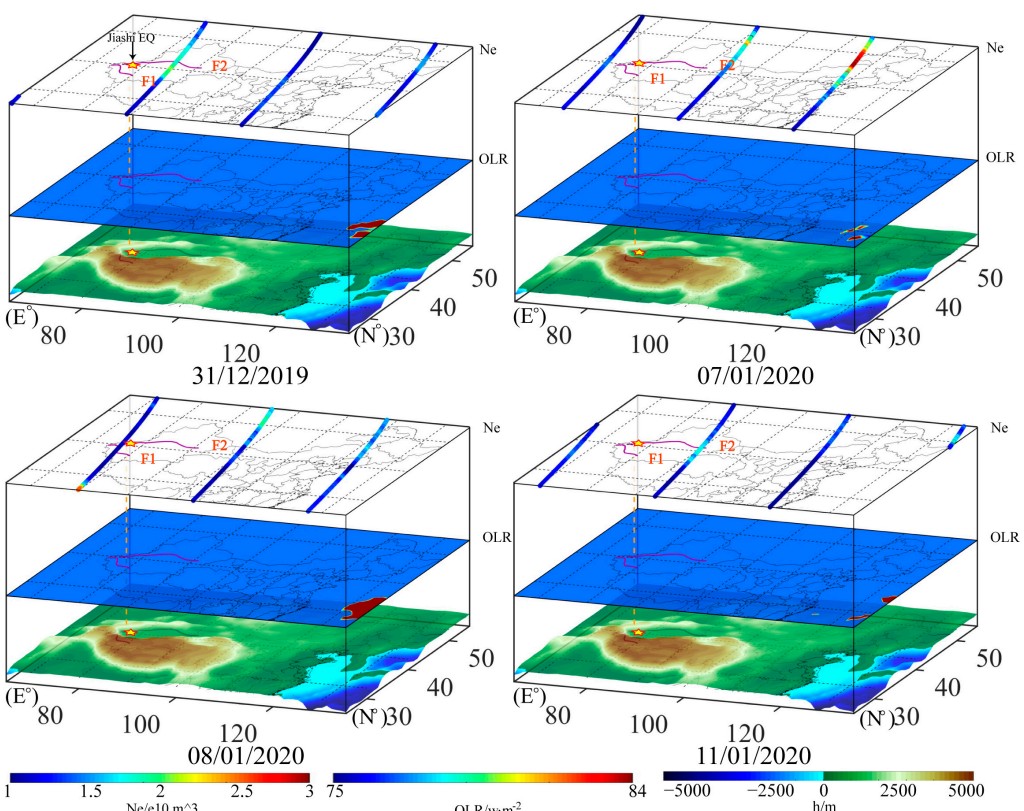

**Figure 5.** Temporal and spatial evolution of OLR and Ne at the epicenter of the Jiashi earthquake (period B). There are no OLR anomalies around the epicenter. Note: The top layer is the CSES orbit map of Ne, the middle layer is the distribution map of the OLR, the bottom layer is the digital elevation model (DEM), the pentagon is the location of the epicenter, F1 is the West Kunlun fault, and F2 denotes the South Tianshan Mountains. The red stars in the top and bottom layers represent the earthquake's epicenter, and the vertical red dashed lines link the epicenters of different layers.

In conclusion, during the earthquake, Ne and OLR exhibited a similar quasi-synchronous evolution process: high initial value → strengthen → peak → weak decay → calm. This phenomenon indicates that during the loading process, the energy input and dissipation, such as rock deformation, fracture, and heat exchange, resulted in changes in the physical temperature of the rock, dielectric constant, and surface occurrence rate, changing infrared and microwave brightness temperatures [53]; moreover, the electrons gathered on the surface caused negatively charged electrons in the atmosphere to move upward. Thus, a quasi-synchronous increase in ionospheric Ne near the epicenter was produced.

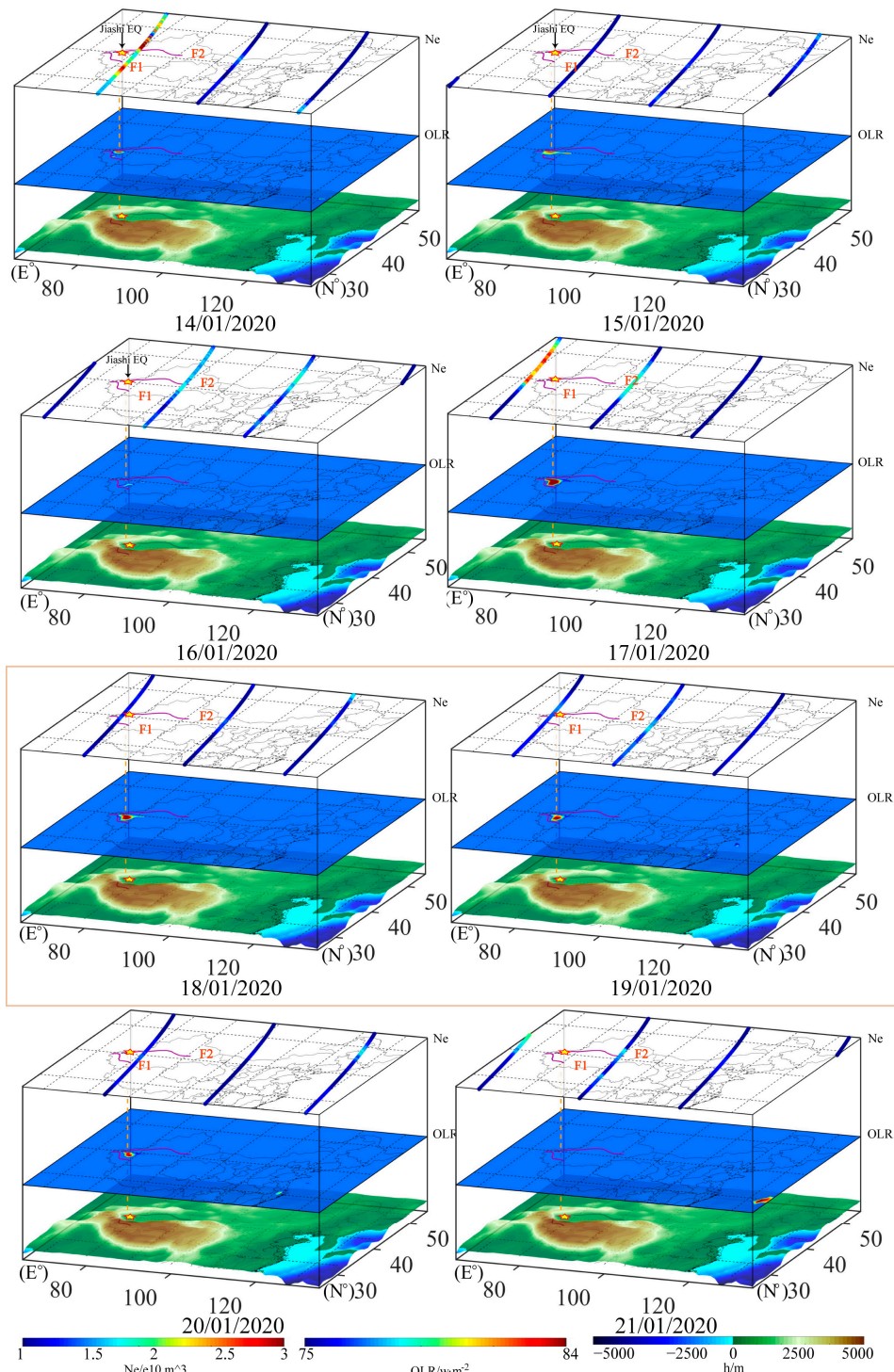

**Figure 6.** Temporal and spatial evolution of OLR and Ne at the epicenter of the Jiashi earthquake (period C). There are OLR anomalies around the epicenter. The evolutions of Ne and OLR during this period and in this region have apparent quasi-synchronous characteristics before and after the earthquake. Note: The top layer is the CSES orbit map of Ne, the middle layer is the distribution map of the OLR, the bottom layer is the digital elevation model (DEM), the pentagon is the location of the epicenter, F1 is the West Kunlun fault, and F2 denotes the South Tianshan Mountains. The red stars in the top and bottom layers represent the earthquake's epicenter, and the vertical red dashed lines link the epicenters of different layers. The red rectangle represents the day of the earthquake.

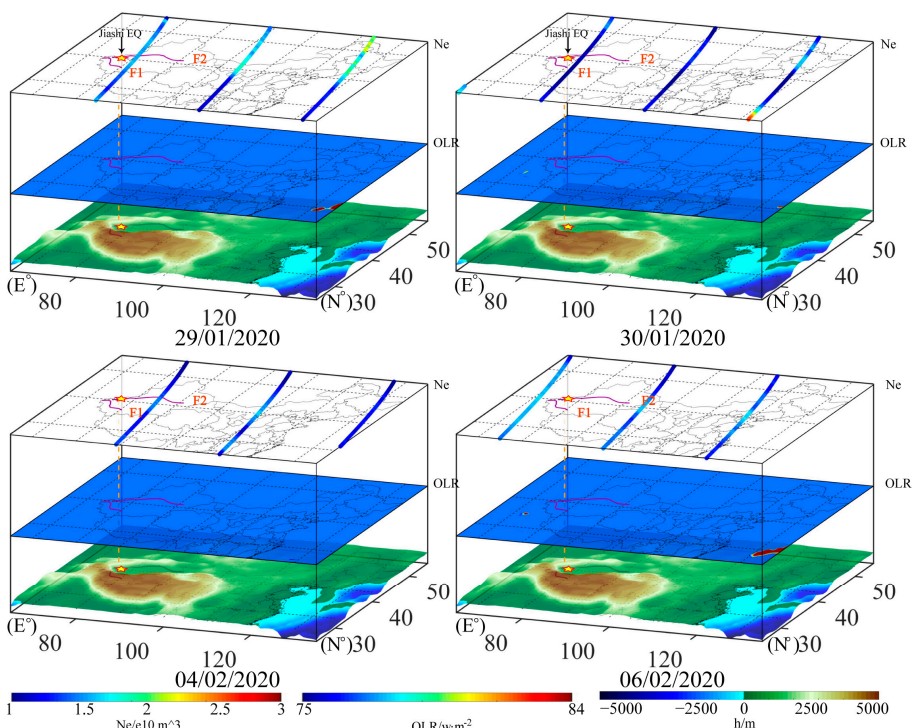

**Figure 7.** Temporal and spatial evolution of OLR and Ne at the epicenter of the Jiashi earthquake (period D). There are no OLR anomalies around the epicenter. Note: The top layer is the CSES orbit map of Ne, the middle layer is the distribution map of the OLR, the bottom layer is the digital elevation model (DEM), the pentagon is the location of the epicenter, F1 is the West Kunlun fault, and F2 denotes the South Tianshan Mountains. The red stars in the top and bottom layers represent the earthquake's epicenter, and the vertical red dashed lines link the epicenters of different layers.

## 5. Discussion

Based on tidal cycles, OLR evolution before the Jiashi earthquake shows that within the wide range region (N 25°–N 55°, E 65°–E 135°), the OLR enhancement phenomenon occurred only near the epicenter, which exhibits uniqueness and high recognition efficiency with respect to the discernibility degree of space. Simultaneously, the OLR evolution process from seismogenic processes to occurrence follows the stress-loading failure law with respect to rock radiation. This phenomenon helps validate the hypothesis that the OLR changes are correlated with earthquakes. The strengthening process of the OLR was synchronized with tidal variations prior to earthquakes; this finding indicates that the tide-generating force changed the process of in situ stress accumulation and imbalance in the structure during the earthquake and played a role in triggering the earthquake. OLR variations may comprise a radiation manifestation of in situ stress, which occurs during the seismic sequence. A comparison of the OLR change process of a similar phase of tide-generating forces before and after the earthquake is shown in Figures 4–7 (A–D periods). There was no obvious radiation enhancement process, and no earthquake occurred during a similar tide-generating force evolution phase within the epicenter area. In summary, this indicates that the precondition for the changing tide-generating forces of celestial bodies relative to triggering an earthquake is as follows: When tectonic stress in the rock at the source of the earthquake accumulates towards a critical state of rock rupture and sliding, the change in the OLR can be used as a remote sensing parameter to monitor the in situ stress state. In addition, it should be noted that several OLR anomalies in the eastern part of China during periods B, C, and D could be connected to other seismic events (≥5.0) that occur in the ocean. Given the several thousand kilometers of distance between the Jiashi seismic sequence and further OLR anomalies, they are not the main object of analysis in this paper.

Figures 4–7 show that Ne and the OLR exhibited similar quasi-synchronous variations during the earthquake. To further illustrate the synchronization of the OLR and Ne, the temporal variations of ΔOLR and ionospheric Ne concentrations over the epicenter of the earthquake in the same plot are shown in Figure 8. According to the empirical equation of the earthquake preparation zone put forward by Dobrovolsky et al. [54], the influential zone of an earthquake is roughly computed via equation $R = 10^{0.43M}$, where R is the diameter of the influential earthquake zone in the km scale, and M is the magnitude of the EQ. For the Ne parameter, the epicenter $\pm 5°$ area was used for analyses. The inter-quantile range method (IQR) was used to analyze the change in Ne [55]. First, the measured data of the orbit around the epicenter were resampled at 0.5°. Second, for each bin of the current orbit, its background can be the median M and IQR of the same-latitude bin values of previous orbits (here, it is 10). Finally, 80% and 20% of the sub-sites are selected as the upper and lower limits.

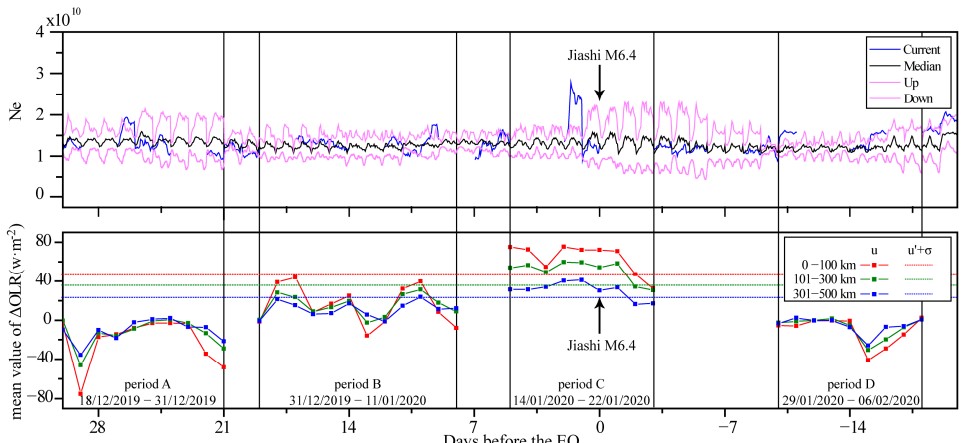

**Figure 8.** Temporal variations in ΔOLR and ionospheric Ne concentration. Top: The background and current observation of Ne. Bottom: The relative changes in OLR.

In the top panel of Figure 8, there are four curves: the current observation (in black color), the median (in blue color), and the upper and lower limits (in cyan color). In the bottom panel of Figure 8, there are three curves: the relative change in the OLR within 0–100 km of the epicenter (in red color), the relative change in the OLR within 101–300 km of the epicenter (in green color), and the relative change in the OLR within 301–500 km of the epicenter (in blue color). Figure 8 shows that both Ne and ΔOLR show significant abnormal increases in period C.

From the top figure of Figure 8, before 18 January, several peaks are from 17 January, and they have a relative positive amplitude of more than 100%. In periods A, B, and C, some peaks are also observed, but the amplitude is very small. In the bottom panel of Figure 8, we found that from period A to period C, the changes in OLR parameters before the earthquake show a gradual upward trend. Period C reaches its peak and presents a continuous high value 5 days before the earthquake, with an average value of about 80 w·m$^{-2}$, and the three ΔOLR is significantly greater than the u + σ. One day after the earthquake, it began to fall back, and from the end of cycle C to cycle D, it exhibited a low-value state, fluctuating around 0 w·m$^{-2}$. In addition, when the OLR parameters did not show significant enhancement changes (such as periods A and D), the amplitude of the change in the three curves did not show significant differences, but when the OLR parameters began to show enhancement changes in periods B and C, the time series curve closest to the epicenter was significantly higher than the other two, and period C's curve was the most obvious, with an enhancement amplitude of 30 w·m$^{-2}$. Although there are weak Ne anomalies in periods A, B, and D, there is no synchronous OLR anomaly, and the Ne anomalies in C cycles are consistent with the OLR. It was further proved

that the synchronization anomaly of infrared and electron concentrations is related to the earthquake. This result is in agreement with some previous studies [56].

The results also indicate that as the TF increases, Earth's stress accumulates at a critical point, and the OLR increases and transfers upward. The theoretical hypothesis underlying the conducted study is that the accumulated electrons on the surface cause the negatively charged electrons in the atmosphere to move upward, leading to an increase in ionospheric Ne near the epicenter, which reveals the homology of seismic stress variations in the spatial coupling process.

## 6. Conclusions

This manuscript presents a case study of a particular earthquake event (the three consequently occurring events) in Jiashi, Xinjiang, which occurred in January 2020. In this study, satellite observation data on outgoing longwave radiation (OLR) and the ionospheric electron concentration (Ne) were analyzed simultaneously, taking into account the variations in the tidal force (TF). The results are presented in the form of three-layered 2D latitudinal–longitudinal graphical schemes. Based on the data analysis, several conclusions were drawn:

(1) The thermal anomalies detected via the tidal force method are related to earthquakes. The periodic variation characteristics of the tidal force not only could provide pre-calculated and determined time background indications for remote sensing short-impending earthquake monitoring, but also add a mechanical basis for thermal anomaly identification before an earthquake, eliminating the uncertainty of the remote sensing earthquake-monitoring conclusions caused by the uncertainty of remote sensing data processing background selections.

(2) Ionospheric disturbance anomalies can occur during the few days before an earthquake. However, due to the wide spatial distribution of anomalies, determining the location of future earthquakes is difficult. Therefore, the comprehensive monitoring of surface radiation and ionospheric electron concentration characteristics is conducive to the study of layer-coupling mechanisms and physical earthquake processes, which are of great significance for earthquake prediction.

(3) The quasi-synchronous variation processes of the TF, OLR, and Ne indicate that the TF changes the stress accumulation and disequilibrium process within the structure, which can trigger the earthquake to some extent. The spatiotemporal variations in the OLR and ionospheric Ne can indirectly reflect the ground's stress.

**Author Contributions:** C.Y., J.C. and W.M. contributed to the conception and design of the study. W.Z. and J.H. organized the database. J.R., performed the geology analysis. C.Y. wrote the first draft of the manuscript. J.C., W.M. and B.S. wrote sections of the manuscript. All authors have read and agreed to the published version of the manuscript.

**Funding:** This work was supported in part by the National Key Research and Development Project (Grant No. 2021YFB3901203), the Earthquake Joint Funds of NSFC (Grant No. U2039205).

**Institutional Review Board Statement:** Not applicable.

**Informed Consent Statement:** Not applicable.

**Data Availability Statement:** The OLR data are available from the following website: https://ftp.cpc.ncep.noaa.gov/-precip/noaa18_1x1/ (accessed on 5 January 2021). The ionospheric electron concentration (Ne) of CSES data utilized in this paper are available from the following website: https://www.leos.ac.cn/ (accessed on 14 January 2021). The Kp and Dst dara are available from the website: https://wdc.kugi.kyoto-u.ac.jp/ (accessed on12 December 2021).

**Acknowledgments:** Thanks to NOAA for the outgoing long-wave Radiation (OLR) data. This work made use of the data from CSES mission, a project funded by China National Space Administration (CNSA) and China Earthquake Administration (CEA). Thanks to the International Space Science Institute (ISSI in Bern, Switzerland and ISSI-BJ in Beijing, China) for supporting International Team

23-583 lead by Dedalo Marchetti and Essam Ghamry. We would also like to thank the reviewers of this paper for their comments and suggestions.

**Conflicts of Interest:** The authors declare no conflict of interest. The funders had no role in the design of the study; in the collection, analyses, or interpretation of data; in the writing of the manuscript; or in the decision to publish the results.

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
