# Peer review of "Quasi-Synchronous Variations in the OLR of NOAA and Ionospheric Ne of CSES of Three Earthquakes in Xinjiang, January 2020"

_atmosphere, doi:10.3390/atmos14121828_

Round 1

Reviewer 1 Report

Comments and Suggestions for Authors

Review of the manuscript  “Quasi-synchronous variation in OLR of NOAA and ionospheric Ne of CSES of three earthquakes in Xinjiang, in January 2020”

Authors: Chen Yu, Jing Cui *, Wanchun Zhang, Weiyu Ma, Jing Ren, Bo Su, Jianping Huang

 The manuscript presents a case study of the particular earthquake event ( the three consequently occurred events) in Jiashi, Xinjiang, in January 2020. In this study, the satellite observation data on the outgoing longwave radiation (OLR) and the ionospheric electron concentration (Ne) are analyzed simultaneously taking into account also the variations of the tidal force (TF). The results are presented in the form of the three-layered 2D latitudinal-longitudinal graphical schemes and discussed in detail in the text of the manuscript.

Based on the data analysis, the authors came to several conclusions:

1.      During the earthquake, Ne and OLR exhibited a similar quasi-synchronous variations;

2.      The analysis results of this earthquake case show that ionospheric disturbance anomalies can occur within a few days before an earthquake.

3.      OLR enhancement phenomenon occurred only near the epicenter;

4.      The strengthening process of OLR was synchronized with tidal force variation prior to earthquakes and that the tide-generating force played a role in triggering the earthquake.

The first conclusion state that the anomalies in OLR and Ne are observed synchronously. This result is in agreement with some previous studies (see, e.g., Ouzounov, D., et al. Atmosphere-ionosphere response to the M9 Tohoku earthquake revealed by multi-instrument space-borne and ground observations: Preliminary results. Earthq Sci 24, 557–564 (2011). https://doi.org/10.1007/s11589-011-0817-z ). The second and third statements also do not appear to be completely new and have already been discussed in some form in the scientific literature. In particular, the results of INSPIRE project (Pulinets S., et al., “Ionosphere Sounding for Pre-seismic Anomalies Identification (INSPIRE): Results of the Project and Perspectives for the Short-Term Earthquake Forecast”, Frontiers in Earth Science, v.9, 2021, DOI=10.3389/feart.2021.610193) have demonstrated that the ionospheric anomalies registered before the strong earthquakes could be used as reliable precursors.

Regarding the last statement, the authors’ conclusion that tidal generating force played a role in the occurrence of the earthquake cannot be justified based on the consideration of a single event. It could just be a coincidence.

In summary, the study presents non-contradictory results, which support already existing models (see, e.g., reference [28] in the manuscript). I would suggest that the authors explicitly indicate in the text, which results of their research are new, and which ones correspond to already known facts. I think that it would be useful to add to the paper one more figure depicting temporal variations of the OLR intensity and the ionospheric electron concentration Ne over the epicenter of the earthquake in the same plot to clear demonstrate their synchronous behavior.

Author Response

Dear reviewer,

Thank you for your comments and suggestions. These comments are valuable and very helpful. We have read thorough comments carefully and have made corrections. We have rewritten the introduction, Method, Discussion and the Conclusions. The language was organized by a language editing company. We uploaded the revised paper and the change-marked-manuscript to the system. Revisions in the text are shown in the change-marked-manuscript. The change-marked-manuscript is after the responses letter. 

Reviewer 2 Report

Comments and Suggestions for Authors

1. The introduction must be improved.  Line 39-40 The definition of an earthquake is confusing here. The meaning of the line is not coming out properly or is difficult to understand.

2. The same thing is applicable for lines 43–44.

3. Line 44 [1] [2] can be [1-2].

4. Line 50. The authors just mentioned the reference and stated that those researchers did the same work for the earthquakes. Which earthquake?.

5. Line 55-67 needs to be rewritten. I cannot understand what the purpose of the paragraph is.

6. Line 69 With the development of space observation technology and an increasing amount of satellite observation data, the disturbances... The sentence is not correct, and it has to be modified. 

Overall, the English editing is essential for the whole manuscript. 

7. Line 93 The link for the earthquake data should be within the bracket before starting the next sentence.

8. Ms5.4 should be Ms 5.4. Similar mistakes are found throughout the manuscript. 

9 KEPINGTAGE is an abbreviation? If it is, then it should be spelled out before using the abbreviation. Also, if it is the name of the fold belt, then it should be Kepingtage. I think.... It should be checked. Because in other papers from the area titled "Kepingtage fault zone,"

10. Similar mistakes are found in the whole manuscript (lines 99–114 and so on).

11. Fig. 1 caption is merged with the paragraph and is also given below Figure 1. Repetition

12. Lines 131–136 need to be rewritten.

13. Figure 3 should indicate what the blue line is. What is the black line? it is very hard to follow up on the diagrams with the description.

14. All the figures must be improved with a descriptive caption. Figure 4-7 is very hard to observe the results.

15. The methodology part is not describing the detailed methodology. It is very difficult to understand what the authors did as an addition. They just use one satellite and make no comparison with other methods. 

16. What is the ionosheric variation with the fault.... It is almost impossible to get information from the figures. 

16. Some places references are written like [1-4], some places [1][2]. It depicts the carelessness of the authors.

18. The font size of the previous paragraph and paragraph (Line 167–186) is not uniform. It gives the impression that the authors are careless about the format of the journal or not serious about the work. 

17. The discussion part must be improved.

In total, although the work is important the data presentation, English and 

Comments on the Quality of English Language

Must be rechecked and improved. 

Author Response

Thank you for your comments and suggestions. These comments are valuable and very helpful. We have read thorough comments carefully and have made corrections. We have rewritten the introduction, Method, Discussion and the Conclusions. The language was organized by a language editing company. We uploaded the revised paper and the change-marked-manuscript to the system. Revisions in the text are shown in the change-marked-manuscript. 

Reviewer 3 Report

Comments and Suggestions for Authors

General Comments:

Overall, the manuscript is very well written and very interesting.  It reports on the new methodology developed.  The authors combined the accurately computed TF values with the measured atmospheric OLR and ionospheric Ne data.  By analyzing their variations of these variables, the authors retrospectively specified the Jiashi earthquake epicenter.  The successful specification was based on the anomalous increased ionospheric Ne density.  The study’s results also demonstrate the theory put forward by the authors that the accumulated electrons on the surface cause the negatively charged electrons in the atmosphere to move upward: leading to an increase in the ionospheric Ne density near the epicenter.  The figures are very detailed and the manuscripts shows that the authors put a lot of effort and work in their study.  Definitely, the manuscript is recommended for publication in its present form.  During the process of proofreading, the authors should make the following corrections (see detailed comments below).

Detailed Comments:

L93: “(http://www.cenc.ac.cn/cenc/dzxx/359679-/index.html)” does not work

L32 and L199: ”w/m2” should read “W/m2” where 2 is superscript.

L197, L200, L246, and L251: ”W/m2” should read “W/m2” where 2 is superscript”

Author Response

Thanks very much for taking your time to review this manuscript and we really appreciate your approval of our article. We have made some modifications based on you and other reviewers in the revised paper. Revisions in the text are shown in the change-marked-manuscript. The change-marked-manuscript is after the responses letter. 

Round 2

Reviewer 1 Report

Comments and Suggestions for Authors

The manuscript has been revised and improved. I am satisfied with the responses to my comments.